# Characterization of Potato Tuber Tissues Using Spatialized MRI T2 Relaxometry

**DOI:** 10.3390/biom13020286

**Published:** 2023-02-02

**Authors:** Guylaine Collewet, Saïd Moussaoui, Stephane Quellec, Ghina Hajjar, Laurent Leport, Maja Musse

**Affiliations:** 1INRAE, UR OPAALE, F-35044 Rennes, France; 2Ecole Centrale Nantes, Nantes Université, F-44321 Nantes, France; 3IGEPP, INRAE, Institut Agro, University of Rennes, F-35653 Rennes, France

**Keywords:** MRI, *T*
_2_, relaxometry, tuber, storage, classification

## Abstract

Magnetic Resonance Imaging is a powerful non-destructive tool in the study of plant tissues. For potato tubers, it greatly assists the study of tissue defects and tissue evolution during storage. This paper describes the MRI analysis of potato tubers with internal defects in their flesh tissue at eight sampling dates from 14 to 33 weeks after harvest. Spatialized multi-exponential $T_{2}\;$relaxometry was used to generate bi-exponential $T_{2}$ maps, coupled with a classification scheme to identify the different $T_{2}\;$homogeneous zones within the tubers. Six classes with statistically different relaxation parameters were identified at each sampling date, allowing the defects and the pith and cortex tissues to be detected. A further distinction could be made between three constitutive elements within the flesh, revealing the heterogeneity of this particular tissue. Relaxation parameters for each class and their evolution during storage were successfully analyzed. The work demonstrated the value of MRI for detailed non-invasive plant tissue characterization.

## 1. Introduction

Magnetic Resonance Imaging (MRI) is a powerful non-destructive tool to study the internal structure and composition of plants and foods [1]. Among the various acquisition protocols available with MRI, $T_{2}$ relaxometry is used to measure the transverse relaxation times using a multi-echo spin echo (MSE) sequence which provides a set of images at a fixed sample rate along an exponential decay signal. $T_{2}$ varies according to the molecular environment and the diffusivity of the water at short distances, providing information on fluid composition (concentration and structure of solutes) and confinement. MRI $T_{2}$ relaxometry has often been employed to estimate mono-exponential relaxation maps of plant tissues [2,3,4], providing averaged information on water protons in different physico-chemical environments. To gain a better insight into the microstructure of plant tissues, the relaxation signal measured in each voxel can also be modeled by a multi-exponential decay, which corresponds to the sum of the signals of the water protons in different environments. These water pools have generally been assumed to reflect subcellular compartmentalization [1] and/or heterogeneity of the tissue [5,6], assuming that the diffusive exchange of water between the compartments which partially averages their relaxation signals can be disregarded. MRI can thus be used to access information on water status and distribution at the cellular and tissue levels. The accurate estimation of multi-relaxation $T_{2}$ parameters at voxel level requires optimal sampling of the relaxation curve and calls for specific algorithms to accommodate the relatively low signal to noise ratio and particular noise distribution. Recently developed methods [7,8] now make it possible to exploit multi-exponential $T_{2}$ information to address issues in both plant biology and food science.

Potato tubers are consumed in fresh and processed form and are used for starch extraction and potato seedling production. Their quality traits are mainly defined by tuber dry matter content and composition, and by the absence of external and internal defects. Factors affecting quality at final harvest include the choice of cultivar along with growth conditions such as soil characteristics, the availability of macro- and micronutrients and climate [9]. The cold storage of potato tubers after harvest is commonly used to extend postharvest shelf life. Loss of potato quality during storage is mostly due to sprouting, weight loss, changes in chemical composition and to the development of defects caused by infections and mechanical impacts at or before harvest. Such defects have a particularly adverse impact on the potato industry as they lead to the loss of commercial revenue. Potato tubers have a complex structure made up of three major tissue types—the cortex, the flesh and the pith—characterized by their own specific histological and metabolic traits [10]. We can therefore expect each tissue type to be affected differently by storage conditions in terms of its evolution over time and the formation of defects.

A recent MRI study has investigated internal defects in potato tubers and their evolution during storage [11]. Multi-exponential $T_{2}$ parameters were estimated in the regions of interest (ROI) corresponding to rust spots present in the flesh and compared to the $T_{2}\;$of the surrounding flesh tissues. In this study, the ROIs were manually defined on $T_{2}$-weighted images and it was assumed that these images reflected differences in the multi-exponential relaxation parameters.

In the present study we applied a recently proposed spatialized multi-exponential $T_{2}\;$relaxometry approach that combines the estimation of multi-exponential $T_{2}\;$at the voxel level with a classification scheme [12] to potato tubers during harvest. The purpose was to cluster automatically, and without a priori, similar voxels according to the multi-exponential relaxation parameters, in order to better distinguish between the constitutive elements of the tissues, including defects, and also to reveal tissue heterogeneity. The evolution of the relaxation parameters of each individual tissue element was monitored over the course of storage.

## 2. Materials and Methods

Details concerning tuber production and storage, and MRI analytical protocols are provided in [11]. Consequently, only a summary of key aspects of the protocols is provided in the following.

### 2.1. Tubers

The potato tubers were harvested from plants of the Rosanna cultivar of *Solanum tuberosum* grown in a greenhouse (IGEPP, Le Rheu, France) under water deficit conditions comprising soil humidity levels equivalent to 20% of field capacity, interspersed with 3 short periods (2–3 days) of rehydration at 33, 46 and 60 days after shoot emergence. After final harvest, following a protocol recommended by the potato industry, tubers were prepared for long-term storage and were then stored at 5°C. At the beginning of the storage period, a set of tubers with defects were visually selected from previously recorded MRI morphological images. Four tubers with large internal rust spots in their flesh tissue were analyzed by spatialized multi-exponential $T_{2}$ relaxometry on 9 occasions over the 5-month storage period. Sampling dates corresponded to 14, 18, 21, 23, 25, 27, 29, 31 and 33 weeks after harvest (WAH).

### 2.2. MRI Acquisition Protocol

Images of potato tubers were recorded on a 1.5 T MRI scanner (Magnetom Avanto, Siemens, Erlangen, Germany) equipped with a circular polarized head array radiofrequency (RF) coil. Tubers were analyzed simultaneously. They were placed in a temperature regulating device within the RF coil, allowing the temperature to be set at the same value as for storage (5 ± 1 °C). Transverse relaxation parameters of the middle section of the tubers were measured using a multi-spin echo (MSE) sequence [7] with the following parameters: 256 echoes with the first echo (TE) equal to the inter-echo spacing, repetition time 10 s, bandwidth 290 Hz/pixel, imaging matrix 160 × 160, field of view 152 mm × 152 mm, slice thickness 5 mm and 2 averages.

The centers of the virtual tuber slices analyzed by MRI were marked with a permanent marker to ensure their position remained the same for the 8 subsequent sampling dates.

### 2.3. Estimation of the Transverse Relaxation Maps

Signal $s\left(x,k\right)$ in voxel x at echo number k was modeled as the sum of decaying exponentials of C components:$$s\left(x,k\right)=\overset{C}{\underset{c=1}{\sum }}A_{0cx}e^{-kTE/T_{2cx}}$$
where $A_{0cx}$ was the amplitude and $T_{2cx}$ the relaxation time of component c in voxel x.

As proposed in [12], the optimal number of components, C, was chosen according to the Bayesian information criterion (BIC). This criterion is commonly used for model order selection since it provides a performance measurement that takes model complexity into account [13]. Figure 1A shows the map of the optimal C values at 14 WAH. The two most frequent values were 2 and 3 (dark and light orange), with very few instances of either 1 (purple) or 4 (white). C = 2 occurred most frequently across all voxels at 14 WAH, and the same result was obtained for all sampling dates, although on some sampling dates greater variability among the voxels was observed for this criterion (Appendix A). This led us to choose this value for the analysis. It was also in accordance with a previous study where analysis of relaxation decay values measured by MRI revealed two major water populations within the potato [11,14]. Thus, for each voxel, two relaxation times, numbered in ascending order, $T_{21x}$ and $T_{22x}$, and the corresponding amplitudes $A_{01x}$ and $A_{02x}$ were estimated.

$T_{21x}$, $T_{22x}$, $A_{01x}$ and $A_{02x}$ were estimated using the method described in [8]. The latter maximizes the likelihood of the data under the Rician noise hypothesis while imposing spatial regularity on the solutions in order to reduce the effects of the noise. This method is iterative and requires initial guess values for each parameter. For all voxels, $T_{21x}$, $T_{22x}$, $A_{01x}$ and $A_{02x}$ were initialized, respectively, at 100 ms, 200 ms, $0.3s\left(x,1\right)$ and $0.7s\left(x,1\right)$ based on a previous study led on the same material, for which $T_{2}$ values estimated over a large region of interest were given [11]. The two parameters to control for spatial regularity were chosen as proposed in [12]. Figure 1B shows the relative fitting error map ***F*** between the measured and the estimated signal at 14 WAH. F was computed as follows for each voxel x, noting $y\left(x,k\right)$ the measured signal in voxel x at echo number k, σ the noise standard deviation and $E_{R}$ the esperance following the Rician distribution
$$F\left(x\right)=\sqrt{\frac{\sum_{k}{\left(y\left(x,k\right)-E_{R}\left(s\left(x,k\right),\sigma \right)\right)}^{2}}{\sum_{k}{\left(y\left(x,k\right)\right)}^{2}}}$$

The relative fitting error did not exceed 0.05 (5%) of the signal. The spatial variation of this error could be explained by the spatial variation of the signal linked to the different kind of tissues. The error was lower for the region where the signal to noise ratio was the highest, which was expected. Similar results were observed for the other sampling dates.

### 2.4. Clustering and Labeling

The estimation step produced four parameter maps. A post-processing scheme that clusters voxels with similar multi-exponential relaxation parameters was applied in order to reduce the complexity of the information, as suggested in [12]. For this, the K-means algorithm which, given a number of classes, clusters voxels with similar parameters [15] was used. The variables used were $T_{21x},\;T_{22x}$ and the amplitude ratio of the longest $T_{2}$ component $R_{02x}=A_{02x}/\left(A_{01x}+A_{02x}\right)$. By using this ratio, rather than the amplitudes, we were able to overcome the spatial variations of the signal linked to the receive coil, which consists in a multiplicative bias, thus corrected by the ratio computation. $T_{21x},\;T_{22x}$ and $R_{02x}$ were divided by their maximum value across all voxels, in order to achieve the same range value for the three parameters. One classification was carried out per sampling date and the four tubers were classified together, thereby increasing the number of voxels in the sample and improving the robustness of the classification.

The selection of the number of classes was based on a compromise between the need to classify the defects from the tuber tissues separately and the need to obtain statistically different classes. Since classification by *K*-means imposes a number of classes and, in no way, guarantees that these classes are statistically different, differences between classes were studied after the classification stage. The differences between the classes were assessed statistically using a univariate approach. We focused on the difference between each pair of classes, which corresponds to the univariate post-hoc analysis of a multivariate analysis. The Kruskal–Wallis test was performed. The alpha level was set to 0.01 and the Bonferonni correction was used as the control for multiple comparisons between each pair of classes.

In order to assess the evolution of the individual classes over time, each class was assigned a color with respect to two characteristics ($T_{22}$ value and proximity to the center). First, the two classes with the highest mean $T_{22}$ values were identified. That closest to the center of the tuber was colored green and the other yellow. The other classes were colored purple, cyan, dark orange and light orange, taking into account only their mean distance from the center of the tuber (from furthest to closest).

## 3. Results

Figure 2 shows $T_{21},\;T_{22}$ and $R_{02}$ maps for 14 WAH. The shortest $T_{2}\;$component varied from around 50 to 220 ms and the longest component varied from 150 to 600 ms, while $R_{02\;}$varied between 55 and 80%. The different structures to be found in the tuber could be observed on these maps, especially the pith, the defects (corresponding to high $T_{2}$ zones) surrounded by the flesh, and the cortex (corresponding to low $T_{21}$ values). Red arrows indicate the location of the defects, determined visually as detailed in [11].

Regarding classification, the number of classes that met the compromise requirements described above was six. Figure 3 depicts the classification results for 14, 23, 25, 27, 31 and 33 WAH. The cortex in purple, the pith in green, and the defects in yellow, were clearly distinguished in all cases, while three classes could be detected in the flesh, including that marked in cyan, which tended to be located along the margin between the cortex and the flesh. This was observed for all of the nine sampling dates except three, 18, 21 and 29 WAH, where some differences were in evidence (see Appendix A). Whereas the defects and the pith were clearly classified for all the sampling dates, in the case of 18, 21 and 29 WAH, the constitutive elements of the flesh and the adjacent cortex were less clearly clustered. For this reason, these three measurement points were excluded from the subsequent analysis to enable assessment of the evolution of all classes over time.

Figure 4 shows the number of voxels for each class by sampling date. Each value corresponds to the sum of the number of voxels for all four tubers. The number of voxels in each class was relatively constant, demonstrating the robustness of the method. Exceptions were observed for 33 WAH, where fewer voxels in the dark orange class were present, and 25 WAH, where more voxels fell into the yellow class. Indeed, for this sampling date, some of the voxels located in the cortex area were classified as defects, as can been seen in Figure 3.

Table 1 lists the *p*-values for each pair of classes and each variable for 14, 23, 25, 27, 31 and 33 WAH. Values >0.01 are shown in red. Two of the three variables $T_{21},\;T_{22}$ and $R_{02}$ were significantly different for each pair of classes, except in the case of the green and yellow (pith and defects) pair, for which only one variable was significantly different, with no significant difference at 14 WAH. Indeed, pith and defects have close $T_{2}$ values. Moreover, the number of voxels for the defect class was small, leading to a relatively high uncertainty on the parameters mean values, leading in turn to a non-significant statistical difference with the pith. Thus, except this particular case, all the different classes, at each sampling date, were statistically different from each other in terms of their bi-exponential $T_{2}$ relaxation parameters.

Figure 5 depicts the scatter plot of $T_{21},\;T_{22}$ and $R_{02}$ values for each voxel with the corresponding color class for 14 WAH. The borders between the purple, light orange, dark orange and cyan classes were relatively straight, suggesting that, in this 3D space, the distribution of these voxels was fairly continuous, unlike that of the voxels of the pith and the defects, which appeared more scattered. The dark and light orange classes had lower $R_{02}$ values compared with the purple class and there was an increasing variation of $T_{22}$ between the light orange, dark orange and cyan classes. The latter observation is interesting, since it corresponds to increasing distance from the tuber center.

In order to provide a representation of the $T_{2}$ values for each class, the distribution of the relaxation times was computed as detailed in [12]: for each $T_{2}$ interval of width 15 ms, the corresponding $R_{0c}$ was added for each component. This sum was divided by the number of voxels in each class to obtain comparable amplitude scales between classes. Figure 6 shows these $T_{2}$ distributions for 14, 23, 25, 27, 31 and 33 WAH. These graphs allow us to observe both mean values and the distribution of the relaxation times for each class. It is somewhat similar to the estimated spectra of a continuous distribution model. The evolution of the mean of each parameter, $T_{21},\;T_{22}$ and $R_{02}$, during storage is shown in Figure 7, with the corresponding standard deviations represented as error bars.

The cortex (purple class) had low $T_{2}$ relaxation values coupled with high $R_{02}$ values. By contrast, the cyan class, located next to the cortex, had higher $T_{2}$ relaxation values and similar $R_{02}$ values. The light orange class had the lowest $T_{2}$ relaxation values, as well as the lowest $R_{02}$ values. The pith and the defects had similar $T_{2}$ values and the defects had lower $R_{02}$ values. With regard to the evolution of relaxation times, $T_{21}$ and $T_{22}$ values, for the defects, and $T_{22}$ values, for the pith, tended to decrease during storage, whereas these parameters were fairly stable for the other classes. Regarding the evolution of $R_{02}$, for all classes other than the cortex, for which no variation was observed, values were relatively stable until 31 WAH, and then tended to rise at 33 WAH. Measurements at 25 WAH differed from those at other sampling points. In addition to the defect detected in the cortex mentioned above, and which probably disturbed the classification, the dark orange class appeared different in the upper right tuber, explaining the different values for this sampling date. The same trends were observed for each tuber separately as can be observed in Appendix A.

## 4. Discussion

In this study, the use of multi-exponential $T_{2}\;$relaxometry was investigated as a means to distinguish defects and the constitutive elements of tuber tissues, and to monitor them over storage. Thanks to a recently published method [8] it was possible to evaluate, at voxel level, two $T_{2}$ relaxation times along with their amplitude ratios. Application of a classification scheme to these parameters allowed for six different classes to be distinguished. Two were attributed to the pith and the defects by noting the positions of the two longest $T_{2}$ values within the tuber, while the four other classes were labeled according to their distance from the tuber center.

The classes’ spatial distribution over time was not consistently recorded for three of the nine sampling dates, leading us to discard these points for the analysis. A possible reason for the discrepancy is that the virtual MRI section may not have always been positioned in exactly the same place, despite all the precautions taken, due to a possible slight variation of the position and angle of the tuber in the container. Other problems may be added to that of positioning but they remain to be elucidated However, for the six remaining sampling dates, it is interesting to note that the classification scheme did not include any spatial information but still led to spatially coherent classes, within each tuber and also over time. The pith, the defects and the cortex were correctly detected and the classification managed to distinguish three different tissues within the flesh. Moreover, the relaxation parameters of these tissues were found to be statistically different. This confirmed that different areas were present within the tubers, with different relaxation parameters.

Regarding the method itself, the choice of the number of classes is open to discussion. The choice was based on a comparison of the results from different numbers of classes, for which some criteria could be considered subjective. Indeed, the process called for user expertise, and is comparable in this respect to other user interventions such as the manual delimitation of ROIs performed on images in previous studies [11,14]. Nevertheless, this novel automated approach made it possible to find non-trivial delimitations between tissues that would not necessarily have been found manually. Additionally, statistical tests were used to verify that the classes did indeed correspond to tissues whose bi-exponential relaxation parameters were actually different.

This method of estimation relies on a model that uses a discrete number of relaxation components. Use of the BIC criterion revealed that two components was the best choice, defined as the most frequently-occurring optimal value across all voxels for all sampling dates. However, for some voxels, the optimal number was smaller or greater. Modeling the signal as the sum of a continuous distribution of $T_{2}$ values would no doubt better reflect reality, but it would complexify the problem. Meanwhile, the two-component model can already make it possible to provide relevant information on the different zones.

The flesh could be divided into three classes that were statistically different for almost all parameters and at all measurement dates, contrasting with the cortex and the pith which appeared clearly as homogeneous regions. Using the classification scheme, we were able to confirm and quantify the heterogeneity in the flesh already observed in a previous study [14]. This property of the flesh could be explained by the fact that flesh tissue results from a continuous process of procambial cell division and growth, while the other two potato tissue-types (i.e., the cortex and the pith) are produced from stolon tissues already in existence [16]. The heterogeneity of the tissue can be assumed to develop during the dynamic phase of tuber growth [10,16,17], given that it was already observed at harvest. This result may reveal differences in growing kinetics driven by sink strength activity and associated with sugar metabolization and starch accumulation within different parts of the flesh tissue. These phenomena are possibly intensified by the effects of water deficit. It should be noted that, within the flesh tissue, these three classes did not display any spatial coherence, contrasting with the coherence to be observed in the different tissues. However, no clear link could be established between the defects and any one specific class within the flesh tissue.

Through its capacity to access information on water status and, consequently, on the latter’s organization at cell level, using the dedicated methods that enable spatialized relaxometry MRI provides a unique perspective on differences in healthy tuber tissues, tissue heterogeneity and the appearance of defects. Interpretation of the relaxation times in terms of tissue structure and composition is beyond the scope of this paper, but some interpretations have already been proposed elsewhere [18,19], Hajjar et al. linked the higher relaxation times observed in pith and rust spots compared with other tissues to their lower starch content [11]. There is scope to improve these interpretations through dedicated studies. More widely, the approach proposed in this study could already find a number of applications in plant science and agronomy. For example, monitoring the appearance times of different defects and their growth and evolution during storage would be of great interest, especially for the number of defects caused by damage induced by machinery or infection at harvest that appear during storage. Further, the classification scheme could be used to monitor in situ tuber growth to gain a better understanding of the spatial variations in this process and the associated development of tuber quality. It could also be used in the identification of the homogeneous areas of tuber tissues of interest for sampling in biocellular studies (e.g., sink-strength quantification, histology, omics parameters).

## 5. Conclusions

Spatialized multi-exponential $T_{2}\;$relaxometry is shown to be a powerful non-invasive tool with the capacity to detect defects and also reveal tissue heterogeneity. Thanks to its generic character, the method described here could in future be used in numerous investigations in plant science designed to increase our understanding of the mechanisms underlying the formation and evolution of tissues.

## Figures and Tables

**Figure 1 biomolecules-13-00286-f001:**
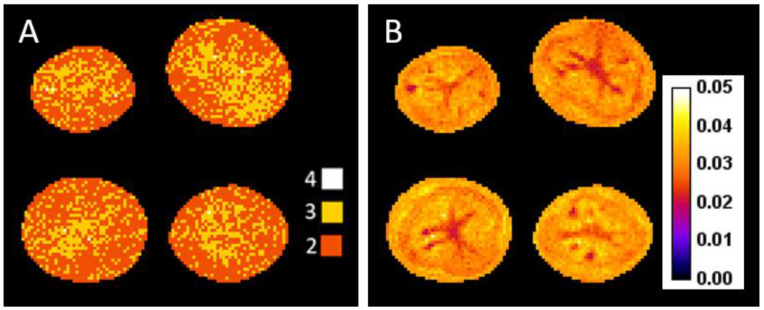
(**A**) Optimal number of model components (2 to 4 as indicated with different color codes) to estimate relaxation parameters for each voxel using the Bayesian Information Criterion, computed for potato images recorded at 14 WAH. (**B**) Relative error between the measured and the estimated signal at 14 WAH.

**Figure 2 biomolecules-13-00286-f002:**
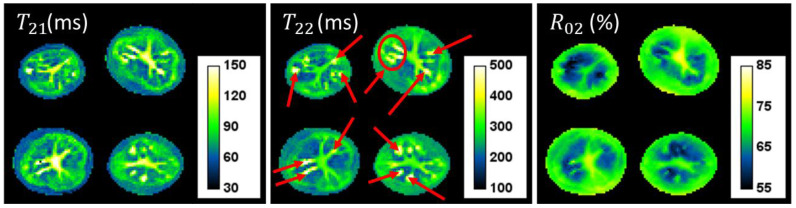
$T_{21},\;T_{22}$ and $R_{02}$ maps for 14 WAH, red arrows indicate the location of the defects.

**Figure 3 biomolecules-13-00286-f003:**
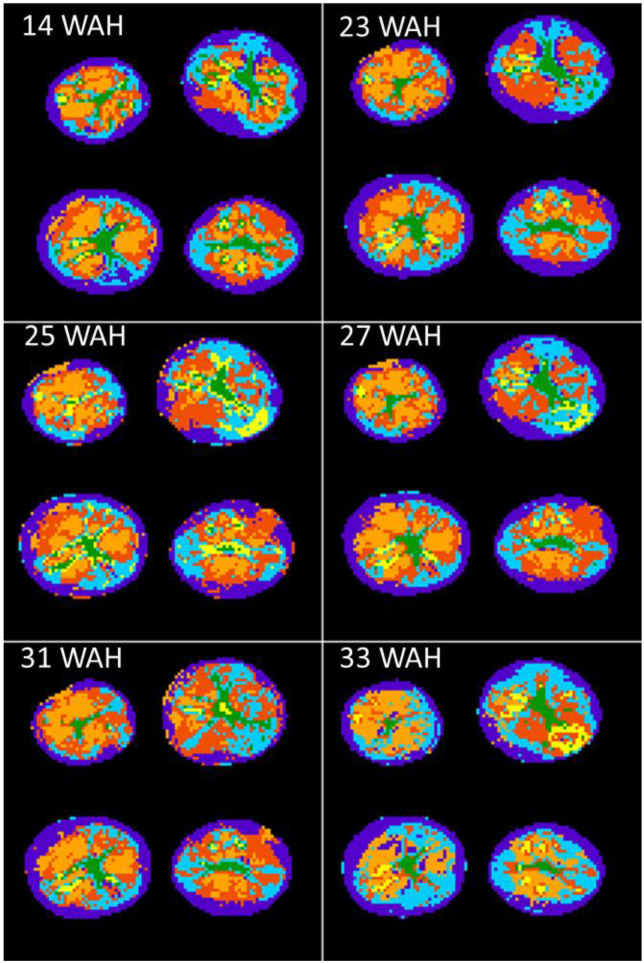
Classification for 14, 23, 25, 27, 31 and 33 WAH. Colors were attributed regarding both $T_{22}$ and spatial information as detailed in the text. On average, purple could be attributed to cortex, green to the pith and yellow to the defects, the three other colors, light and dark orange and cyan to the flesh.

**Figure 4 biomolecules-13-00286-f004:**
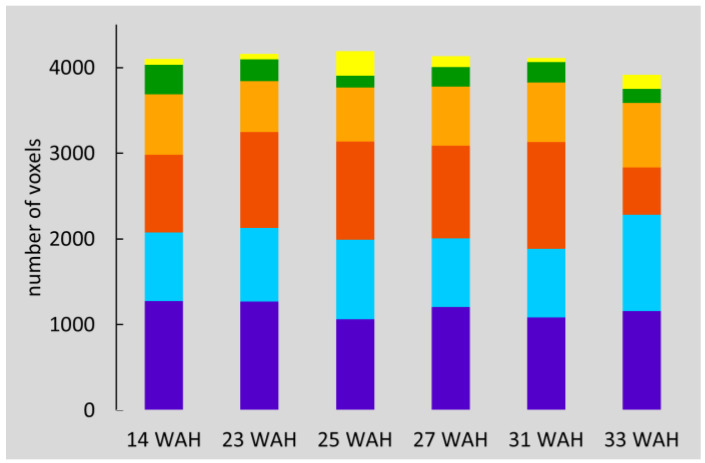
Number of voxels for the four tubers grouped by class and sampling date. Colors were attributed regarding both $T_{22}$ and spatial information as detailed in the text. On average, purple could be attributed to cortex, green to the pith and yellow to the defects, the three other colors, light and dark orange and cyan, to the flesh.

**Figure 5 biomolecules-13-00286-f005:**
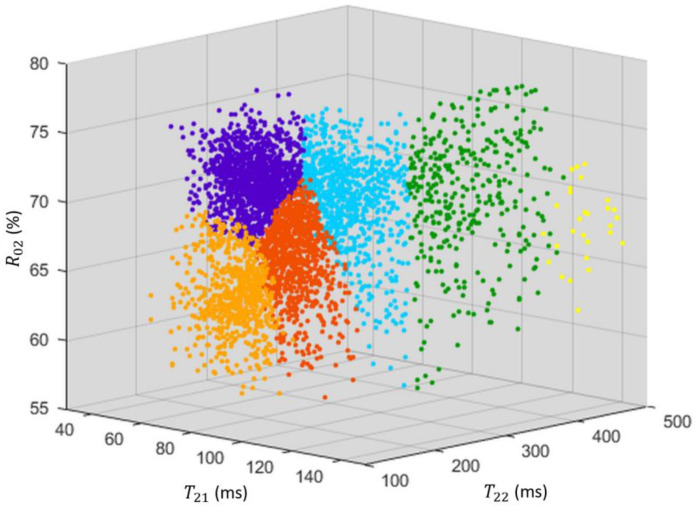
$T_{21},\;T_{22}$ and $R_{02}\;$ for each voxel with the corresponding color class for WAH 14. Colors were attributed regarding both $T_{22}$ and spatial information as detailed in the text. On average, purple could be attributed to cortex, green to the pith and yellow to the defects, the three other colors, light and dark orange and cyan, to the flesh.

**Figure 6 biomolecules-13-00286-f006:**
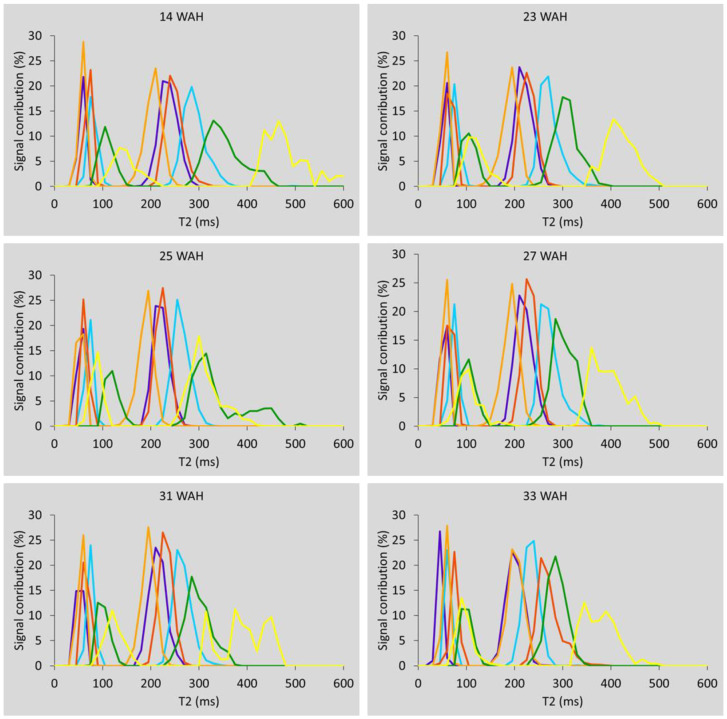
$T_{2}$ distribution for 14, 23, 25, 27, 31 and 33 WAH. Colors were attributed regarding both $T_{22}$ and spatial information as detailed in the text. On average, purple could be attributed to cortex, green to the pith and yellow to the defects, the three other colors, light and dark orange and cyan, to the flesh.

**Figure 7 biomolecules-13-00286-f007:**
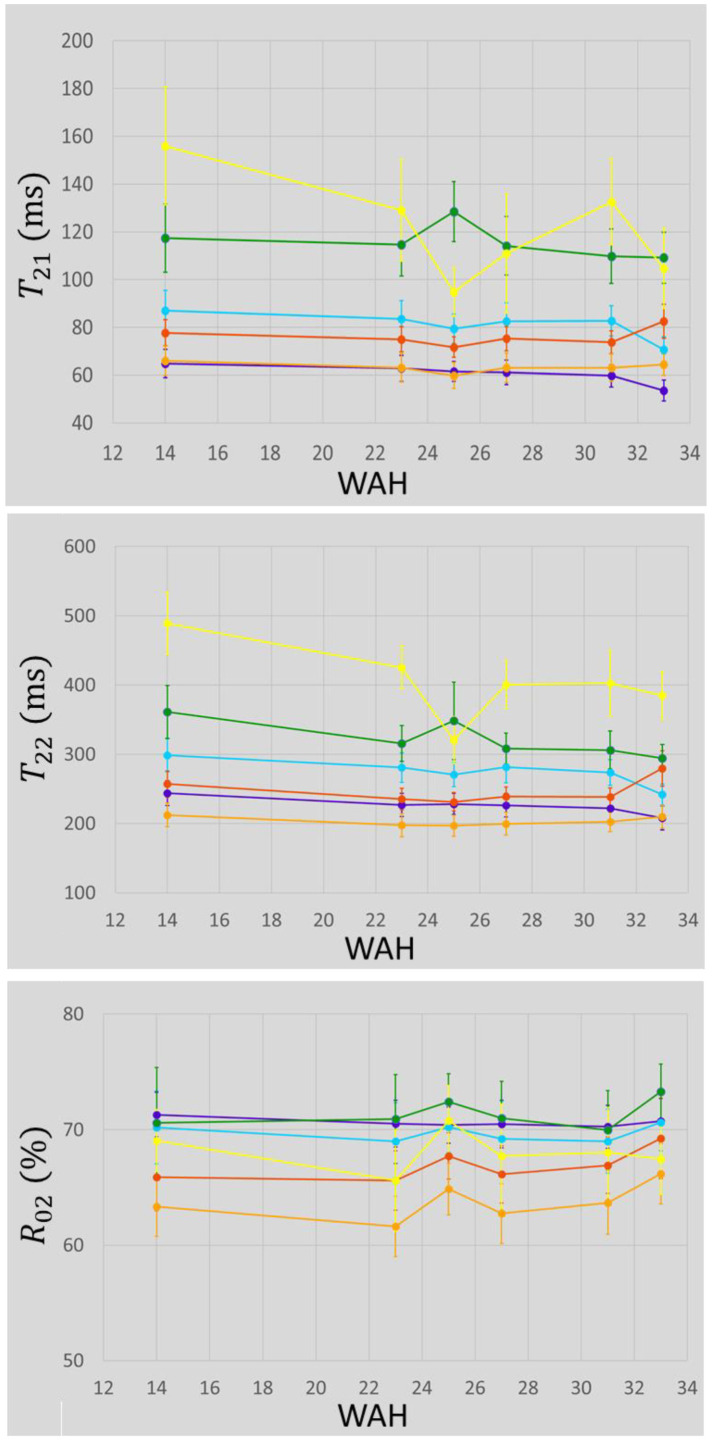
Evolution of the mean values of $T_{21},\;T_{22}$ and $R_{02}$ over time. Error bars correspond to standard deviations. Colors were attributed regarding both $T_{22}$ and spatial information as detailed in the text. On average, purple could be attributed to cortex, green to the pith and yellow to the defects, the three other colors, light and dark orange and cyan, to the flesh.

**Table 1 biomolecules-13-00286-t001:** *p*-values for each pair of classes and each variable for 14, 23, 25, 27, 31 and 33 WAH. Values >0.01 are shown in red. Colors were attributed regarding both $T_{22}$ and spatial information as detailed in the text. On average, purple could be attributed to cortex, green to the pith and yellow to the defects, the three other colors, light and dark orange and cyan, to the flesh.

Classes		14 WAH			23 WAH			25 WAH		
A	B	$T_{21}$	$T_{22}$	$R_{02}$	$T_{21}$	$T_{22}$	$R_{02}$	$T_{21}$	$T_{22}$	$R_{02}$
		0	5 × 10^−206^	3 × 10^−9^	0	1 × 10^−256^	2 × 10^−21^	0	1 × 10^−193^	1
		6 × 10^−174^	4 × 10^−21^	7 × 10^−233^	5 × 10^−208^	3 × 10^−14^	8 × 10^−241^	2 × 10^−166^	2 × 10^−2^	1 × 10^−136^
		1 × 10^−1^	7 × 10^−82^	0	1	1 × 10^−76^	0	3 × 10^−1^	3 × 10^−88^	4 × 10^−282^
		0	9 × 10^−211^	2 × 10^−3^	8 × 10^−288^	5 × 10^−164^	1	1 × 10^−184^	1 × 10^−100^	3 × 10^−10^
		2 × 10^−93^	9 × 10^−63^	3 × 10^−6^	2 × 10^−86^	2 × 10^−59^	2 × 10^−19^	4 × 10^−274^	1 × 10^−170^	5 × 10^−1^
		9 × 10^−30^	2 × 10^−86^	3 × 10^−119^	2 × 10^−36^	2 × 10^−150^	6 × 10^−92^	1 × 10^−47^	3 × 10^−161^	1 × 10^−110^
		5 × 10^−226^	0	2 × 10^−214^	4 × 10^−246^	0	1 × 10^−206^	0	0	5 × 10^−247^
		1 × 10^−20^	2 × 10^−13^	1	8 × 10^−19^	3 × 10^−6^	4 × 10^−9^	1 × 10^−17^	6 × 10^−10^	1 × 10^−12^
		1 × 10^−8^	2 × 10^−7^	7 × 10^−2^	6 × 10^−7^	6 × 10^−5^	5 × 10^−8^	5 × 10^−14^	2 × 10^−13^	2 × 10^−2^
		2 × 10^−105^	6 × 10^−153^	5 × 10^−21^	1 × 10^−124^	2 × 10^−133^	1 × 10^−43^	6 × 10^−149^	4 × 10^−117^	7 × 10^−50^
		2 × 10^−76^	6 × 10^−117^	2 × 10^−77^	3 × 10^−69^	5 × 10^−110^	1 × 10^−86^	2 × 10^−57^	1 × 10^−87^	8 × 10^−76^
		2 × 10^−25^	2 × 10^−39^	1 × 10^−8^	2 × 10^−22^	2 × 10^−42^	1	2 × 10^−70^	1 × 10^−148^	1 × 10^−73^
		2 × 10^−263^	0	3 × 10^−142^	7 × 10^−232^	3 × 10^−306^	3 × 10^−170^	5 × 10^−187^	1 × 10^−214^	7 × 10^−145^
		2 × 10^−81^	8 × 10^−123^	5 × 10^−22^	1 × 10^−78^	7 × 10^−116^	8 × 10^−10^	6 × 10^−264^	0	2 × 10^−164^
		1	1	3 × 10^−2^	1	1	5 × 10^−17^	1 × 10^−1^	1	9 × 10^−5^
Classes		27 WAH			31 WAH			33 WAH		
A	B	$T_{21}$	$T_{22}$	$R_{02}$	$T_{21}$	$T_{22}$	$R_{02}$	$T_{21}$	$T_{22}$	$R_{02}$
		0	7 × 10^−252^	7 × 10^−16^	0	7 × 10^−283^	8 × 10^−17^	5 × 10^−304^	1 × 10^−187^	1
		1 × 10^−238^	1 × 10^−29^	1 × 10^−201^	1 × 10^−254^	4 × 10^−57^	3 × 10^−156^	0	0	4 × 10^−18^
		3 × 10^−4^	5 × 10^−68^	0	2 × 10^−11^	1 × 10^−43^	0	2 × 10^−92^	1	1 × 10^−206^
		1 × 10^−272^	2 × 10^−143^	1	1 × 10^−306^	1 × 10^−178^	7 × 10^−1^	2 × 10^−241^	2 × 10^−150^	3 × 10^−18^
		2 × 10^−146^	5 × 10^−115^	1 × 10^−15^	3 × 10^−77^	8 × 10^−52^	1 × 10^−6^	5 × 10^−229^	1 × 10^−194^	1 × 10^−35^
		2 × 10^−24^	6 × 10^−115^	1 × 10^−79^	3 × 10^−49^	4 × 10^−109^	1 × 10^−53^	9 × 10^−44^	8 × 10^−50^	6 × 10^−14^
		2 × 10^−224^	0	1 × 10^−207^	1 × 10^−248^	0	5 × 10^−183^	2 × 10^−36^	3 × 10^−134^	2 × 10^−188^
		7 × 10^−19^	8 × 10^−4^	8 × 10^−8^	5 × 10^−14^	2 × 10^−5^	6 × 10^−3^	4 × 10^−47^	8 × 10^−30^	7 × 10^−21^
		3 × 10^−7^	2 × 10^−8^	6 × 10^−4^	1 × 10^−4^	1 × 10^−3^	1 × 10^−1^	2 × 10^−41^	1 × 10^−50^	7 × 10^−32^
		3 × 10^−128^	5 × 10^−161^	9 × 10^−48^	9 × 10^−113^	4 × 10^−178^	5 × 10^−62^	4 × 10^−124^	5 × 10^−263^	3 × 10^−66^
		2 × 10^−57^	1 × 10^−77^	2 × 10^−72^	7 × 10^−70^	4 × 10^−83^	1 × 10^−41^	5 × 10^−7^	5 × 10^−1^	3 × 10^−41^
		1 × 10^−26^	2 × 10^−68^	2 × 10^−6^	8 × 10^−19^	4 × 10^−26^	5 × 10^−1^	5 × 10^−5^	7 × 10^−7^	4 × 10^−10^
		1 × 10^−206^	9 × 10^−270^	2 × 10^−155^	1 × 10^−211^	2 × 10^−288^	4 × 10^−121^	5 × 10^−98^	1 × 10^−133^	2 × 10^−140^
		1 × 10^−114^	7 × 10^−208^	3 × 10^−35^	5 × 10^−58^	2 × 10^−85^	2 × 10^−12^	4 × 10^−90^	7 × 10^−174^	2 × 10^−4^
		1	2 × 10^−1^	9 × 10^−13^	1	1	6 × 10^−4^	1	1 × 10^−1^	6 × 10^−59^

## Data Availability

The MRI data presented in this study are openly available at: https://doi.org/10.57745/DR2GSS (accessed on 29 January 2023).

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
