# Peer review of "Characterization of Potato Tuber Tissues Using Spatialized MRI T2 Relaxometry"

_biomolecules, 2023, doi:10.3390/biom13020286_

Round 1

Author Response

We thank you for your comments and constructive remarks, which allowed us to improve our manuscript.

  • In page 3, line 115, Fig S1 is not found in the manuscript.

We apologize for this mistake. The supplementary data had been uploaded to the site upon submission but probably not in the right way.

  • What could be the cause or any probable explanation for the exception that had been mentioned in line 194 (Fig 5).

Defects and pith have close T2 values. Moreover, the number of voxels for the defect class was small, leading to a relatively high uncertainty on the parameters mean values, leading in turn to a non- significant statistical difference with the pith. This explanation has been added in the text

  • In the discussion part, the number of references is very less. If possible, the result should be compared with other similar works by citing more references.

    Some references have been added.

Reviewer 3 Report

This work shows how MRI multi-exponential transverse relaxation (T2) relaxometry can be employed to investigate tissue heterogeneity and defects of potato tubers in a non-destructive manner. Even though the authors monitored potato tubers stored at 5ºC at nine different times after harvest (between 14 and 33 weeks), three of them where later excluded for the analysis of evolution over time. Bi-exponential T2 maps proved suitable for this study, which were consistent with previous studies that revealed two major water populations in the tissues. The shortest T2 components ranged between 50-200 ms whereas the longest T2 components were between 150 - 600 ms, and the amplitude rate of the latter varied between 55-80%. Six clases of homogenous zones within the tuber, each with characteristic relaxation parameters, could readily be identified and assigned to the cortex, the pith, the defects, and three different flesh constituents; the differences between classes were assessed statistically using a univariate approach. This approach allowed to find delimitations between tissues that otherwise could be difficult to find manually. Differences during storage were observed mainly in the flesh region, whereas the cortex and the pith remained as homogenous regions.

The manuscript is well written and the data is presented in a clear and well structured manner. The introduction to the topic is clear and the references are adequate.The methodology and the results are clearly described. The conclusions are supported by the data. The figures have good quality and are pertinent; however, some figure legends can be improved for clarity (see comments below). 

I believe this is a nice work that contains enough original material, and the procedure described herein could be further extended to the study other plant systems. I consider this work may be of interest for readers of this journal, thus I suggest accepting it for publication with minor revisions (see below).

Comments:

In the plots of Figure 6 and 7 there are significant differences in the measurements carried out at 25 weeks after harvest. Is this "average" pattern observed for each of the four tubers? Could the authors please supply the plots for each individual tuber as supporting information and improve the discussion about this aspect. 

Figure S1 and S2 are mentioned in lines 115 and 174 but not supplied with the manuscript. Please add this information.

Figure 1 legend. After “components” add “(1 to 4 as indicated with different color codes)” 

Figures 3, 4, 5, 6 and 7 legends. Please indicate what the different colors are assigned to (cortex, pith, etc).

Figure 5. Please add the corresponding units "(ms)" after T21 and T22 in the axes labels.

Figure 6. Please add the corresponding ticks in both axes. 

Figure 7. I recommend changing "WAH" for "time after harvest (weeks)"

Author Response

We thank you for your comments and constructive remarks, which allowed us to improve our manuscript.

  • In the plots of Figure 6 and 7 there are significant differences in the measurements carried out at 25 weeks after harvest. Is this "average" pattern observed for each of the four tubers? Could the authors please supply the plots for each individual tuber as supporting information and improve the discussion about this aspect.

We have put the graphs for each tuber individually in additional data. They all follow the same pattern. This pattern is perhaps due to a variation in the classification linked to the appearance of a particular zone in the potato at the top right which has not yet been elucidated. We have added this discussion in the text.

  • Figure S1 and S2 are mentioned in lines 115 and 174 but not supplied with the manuscript. Please add this information.

We apologize for this mistake. The supplementary data had been uploaded to the site upon submission but probably not in the right way.

  • Figure 1 legend. After “components” add “(1 to 4 as indicated with different color codes)”
  • Figures 3, 4, 5, 6 and 7 legends. Please indicate what the different colors are assigned to (cortex, pith, etc).
  • Figure 5. Please add the corresponding units "(ms)" after T21 and T22 in the axes labels.
  • Figure 6. Please add the corresponding ticks in both axes.

All these recommandations have been taken into account in the new version

  • Figure 7. I recommend changing "WAH" for "time after harvest (weeks)"

We understand this recommendation. However, we have chosen to keep WAH for practical reasons, in particular when citing the day of measurement in the text and in all the figures. In addition, we have already encountered in publications on plant monitoring studies, the expressions DAF (Day After Flourish) or DAS (Day After Sawing)

Reviewer 4 Report

The manuscript by G. Collewet et al. describes the Characterization of potato tuber tissues using spatialized MRI T2 relaxometry. In particular, a spatialized multi-exponential ?2 relaxometry approach, recently proposed by the authors, is applied to generate bi-exponential ?2 maps, coupled with a classification scheme to identify the different T2 homogeneous zones within the tubers.

The body of work presented here is appropriate for the Special Issue: Recent Advances within Plant Spectroscopy and the authors seem to have a good understanding of the subjected matter.

Some points should be addressed before acceptance:

The supplementary Materials (Figure S1 and Figure S2) are not available, or were not entered correctly by the authors

Figure legends (Figure 2, 3, 4, 5, 6) should be more descriptive for better reader understanding

All references must be corrected according to the Biomolecules “template”

If possible, please expand the bibliography by adding some comparisons with similar research

Author Response

We thank you for your comments and constructive remarks, which allowed us to improve our manuscript.

  • The supplementary Materials (Figure S1 and Figure S2) are not available, or were not entered correctly by the authors

We apologize for this mistake. The supplementary data had been uploaded to the site upon submission but probably not in the right way.

  • Figure legends (Figure 2, 3, 4, 5, 6) should be more descriptive for better reader understanding
  • All references must be corrected according to the Biomolecules “template”

These recommendations have been taken into account in the new version

  • If possible, please expand the bibliography by adding some comparisons with similar research

Some references have been added to the discussion.